# Treatment with the Olive Secoiridoid Oleacein Protects against the Intestinal Alterations Associated with EAE

**DOI:** 10.3390/ijms24054977

**Published:** 2023-03-04

**Authors:** Beatriz Gutiérrez-Miranda, Isabel Gallardo, Eleni Melliou, Isabel Cabero, Yolanda Álvarez, Marta Hernández, Prokopios Magiatis, Marita Hernández, María Luisa Nieto

**Affiliations:** 1Instituto de Biomedicina y Genética Molecular de Valladolid (IBGM-CSIC/UVa), 47003 Valladolid, Spain; 2Laboratory of Pharmacognosy and Natural Products Chemistry, Department of Pharmacy, National and Kapodistrian University of Athens, Panepistimiopolis Zografou, 15771 Athens, Greece; 3Laboratorio de Biología Molecular y Microbiología, Instituto Tecnológico Agrario de Castilla y León (ITACyL), 47071 Valladolid, Spain

**Keywords:** multiple sclerosis, EAE, oleacein, intestinal permeability, inflammation

## Abstract

Multiple sclerosis (MS) is a CNS inflammatory demyelinating disease. Recent investigations highlight the gut-brain axis as a communication network with crucial implications in neurological diseases. Thus, disrupted intestinal integrity allows the translocation of luminal molecules into systemic circulation, promoting systemic/brain immune-inflammatory responses. In both, MS and its preclinical model, the experimental autoimmune encephalomyelitis (EAE) gastrointestinal symptoms including “leaky gut” have been reported. Oleacein (OLE), a phenolic compound from extra virgin olive oil or olive leaves, harbors a wide range of therapeutic properties. Previously, we showed OLE effectiveness preventing motor defects and inflammatory damage of CNS tissues on EAE mice. The current studies examine its potential protective effects on intestinal barrier dysfunction using MOG_35-55_-induced EAE in C57BL/6 mice. OLE decreased EAE-induced inflammation and oxidative stress in the intestine, preventing tissue injury and permeability alterations. OLE protected from EAE-induced superoxide anion and accumulation of protein and lipid oxidation products in colon, also enhancing its antioxidant capacity. These effects were accompanied by reduced colonic IL-1β and TNFα levels in OLE-treated EAE mice, whereas the immunoregulatory cytokines IL-25 and IL-33 remained unchanged. Moreover, OLE protected the mucin-containing goblet cells in colon and the serum levels of iFABP and sCD14, markers that reflect loss of intestinal epithelial barrier integrity and low-grade systemic inflammation, were significantly reduced. These effects on intestinal permeability did not draw significant differences on the abundance and diversity of gut microbiota. However, OLE induced an EAE-independent raise in the abundance of *Akkermansiaceae* family. Consistently, using Caco-2 cells as an in vitro model, we confirmed that OLE protected against intestinal barrier dysfunction induced by harmful mediators present in both EAE and MS. This study proves that the protective effect of OLE in EAE also involves normalizing the gut alterations associated to the disease.

## 1. Introduction

Multiple sclerosis (MS) is an immune-mediated, chronic neurodegenerative disease characterized by a persistent inflammatory and oxidative state that leads to axon demyelination and neuroaxonal degeneration [1,2]. The etiology is not well understood, but arises from a complex interplay between genetics and environmental factors [3]. The heterogeneity of the MS symptoms includes muscle weakness, spasticity, paralysis, blurred vision, and gastrointestinal problems, among others [4]. Bladder and bowel symptoms have been rated as the third most important after spasticity and incoordination. Alterations of gut-derived products, intestinal permeability, and enteric nervous system functions have been described in MS patients, and the gut-brain axis is being considered as a key player in MS pathogenesis [5,6]. Thus, intestinal mucosal barrier breakdown will allow microorganisms, pathogens, and potentially large antigenic molecules to pass through; it will destroy the immune homeostasis and, subsequently, trigger systemic inflammatory response, and participate in the development of autoimmune diseases in the final [7].

At present, no cure for MS is known, and current therapies are directed towards modulation of the immune response to reduce the severity and relapses of the disease. However, given the increasing evidence that support oxidative stress as an important component in the pathogenesis of MS, other treatment regimens, including antioxidants, might confer beneficial effects [2,8]. Furthermore, looking for non-canonical targets may guide the field towards future therapeutic approaches in MS.

Experimental autoimmune encephalomyelitis (EAE) induced with a myelin oligodendrocyte glycoprotein (MOG) peptide is one of the most popular experimental models used for studying MS [9]. The MOG_35-55_-induced EAE in rodents closely resembles the clinical and immunopathological features of the human disease, including some intestinal alterations [10,11].

Oleacein (OLE) is one of the main secoiridoids of extra virgin olive oil (EVOO). OLE is released during the mechanical extraction process by the action of the olive fruit enzymes acting on precursor molecules such as oleuropein. Many of the beneficial health properties of EVOO have been attributed to a high content of monounsaturated acids, as well as to other minor components, among which phenolic alcohols and secoiridoid derivatives such as the OLE are found [12,13]. Although OLE does not exist in the intact olive leaves, it can be very easily produced from them, as has recently been shown [14]. OLE can be produced during the extraction procedure by the combined action of oleuropein glucosidase and demethylase, which are present in the leaves, on oleuropein, which is the most abundant secoiridoid in the olive leaves. The production of OLE from olive leaves using a large-scale and affordable method of selective extraction has offered easy access to this molecule for further investigation and also for development as an ingredient of food supplements or potential new drugs. OLE has demonstrated to possess antioxidant, anti-inflammatory, anti-proliferative and immunomodulatory bioactivities that are partially responsible for the beneficial effects of EVOO on human health [13,14,15,16,17,18]. Moreover, in vivo administration of OLE did not exhibit signs of toxicity [19]. The wide spectrum of its biological activities includes cardioprotective, antimicrobial, neuroprotective, and anti-cancer effects [15,20,21]. In a recent preclinical study, we observed that, by targeting immune–inflammatory and oxidative responses, OLE improved clinical signs and motor deficits of EAE mice, suggesting a protective role of this secoiridoid against this neurodegenerative disorder. However, the effect of OLE on the intestinal alterations linked to MS/EAE has not been addressed. In this study, we focused on EAE-intestinal dysfunction and we unraveled the impact of OLE treatment on gut barrier protection.

## 2. Results

### 2.1. OLE Treatment Protected against EAE-Induced Intestinal Mucosal Barrier Damage in Mice

In previous research, we demonstrated that OLE administration to EAE mice protected CNS tissues from inflammatory damage and was sufficient to ameliorate the classical EAE neurological signs. Herein, we addressed the OLE effect on EAE-associated gut intestinal dysfunction. The study was performed on day 24 after EAE induction (acute phase of the disease): Mice of the untreated-EAE group showed one-sided hind limb paralyses (clinical score 2), at minimum, and mice in the OLE-treated EAE group showed an inability to curl the distal end of the tail (clinical score 0.5) (Figure 1B and Appendix A). EAE disease severity was quantified using a standard numerical scale as described in Methods [22]. Although OLE treatment significantly reduced the disease severity in EAE mice, no major changes were found on disease incidence (Untreated EAE, 10/10; OLE-treated EAE mice, 9/10). 

Firstly, we performed a macroscopic inspection of the intestine. We did not observe significant differences in the ratio colon length/body weight among mice of the different experimental groups (Figure 1C,D). Besides, cecal examination showed that the full cecal weight, as well as the ratio full cecum weight/body weight (cecal index), were higher in EAE mice, and OLE treatment prevented this increase (Figure 1E,F); this change may warrant further investigation. 

Next, to investigate the impact of OLE treatment on intestinal barrier function on EAE mice, we evaluated ex vivo the intestinal permeability to 40 kDa FITC-labelled dextran in a non-everted gut sac model using two different segments of intestine: colon and ileum (Figure 2A,B). Colonic and ileal sacs from EAE mice tissues exhibited an increased paracellular permeability when compared to those of the control group (*p* < 0.001). However, intestinal sacs from OLE-treated EAE mice displayed a reduced FD40 passage demonstrating that OLE was able to preserve the gut barrier function.

In addition, we evaluated surrogate serological markers of impaired intestinal permeability and microbial translocation; the intestinal fatty acid-binding protein (iFABP) and sCD14. As shown in Figure 2C, EAE induction significantly increased the serum levels of iFABP and sCD14 compared with healthy control mice (*p* < 0.001), whereas OLE treatment significantly attenuated this response (*p* < 0.01).

Next, AB-PAS staining was conducted to determine changes in mucins content in the colon of mice of the different experimental groups. As shown in Figure 3A, OLE treatment prevented the significant decrease in the overall AB/PAS staining detected in colon sections from untreated-EAE mice. Although we observed a significant reduction in the expression of both acidic and neutral mucins in colon of EAE mice, the ratio acidic/neutral mucin species between the healthy control and EAE mice kept constant at 2:1. 

The expression levels of galectin-3 (Gal-3), a protein linked to mucin expression, were decreased in colon from EAE mice, compared with healthy control mice (*p* < 0.01), whereas treatment with OLE prevented this reduction, keeping values similar to those of the control group (Figure 3B). In contrast, higher Gal-3 levels were found in the serum of EAE mice when compared with the control group, and OLE administration protected them from this rise (Figure 3C). In addition, the expression levels of the glial-derived neurotrophic factor (GDNF), a novel regulator of the intestinal epithelial barrier function, was diminished in both colon and serum of EAE mice, and OLE treatment abolished this reduction (Figure 3B,C) [23]. 

It is worth noting that OLE administration to the control group did not significantly affect any of the above studied parameters.

### 2.2. OLE Treatment Reduced Colon Levels of Inflammatory Markers in EAE Mice

We also examined parameters of intestinal inflammation in the different experimental groups. We found that OLE significantly reduced the levels of the pro-inflammatory cytokines TNFα and IL-1β, which were observed up-regulated in colon tissue from EAE mice (Figure 4). Additionally, the expression levels of the two potent type-2 inducing cytokines IL-33 and IL-25 were down-regulated in colon from EAE mice, compared to healthy control mice; and OLE treatment protected against this decrease (Figure 4).

### 2.3. OLE Treatment Reduced Colon Levels of Oxidative Stress in EAE Mice

To measure intestinal stress injuries, superoxide anion (O_2_·^−^) accumulation was measured in situ using the DHE stain (Figure 5A). We observed elevated red fluorescence in colon sections from EAE mice compared to control mice (*p* < 0.001), which indicated excess superoxide levels. In contrast, OLE treatment prevented these increases (*p* < 0.001).

Other parameters which indirectly reflect the oxidative extent of cells/tissues, the levels of MDA (as a lipid peroxidation marker) and AOPP (as an oxidative modified proteins marker) were also found significantly augmented in colon from EAE mice (*p* < 0.001; Figure 5B,D), whereas FRAP levels (as an indicator of non-enzymatic antioxidant status) were significantly decreased when compared with the healthy control group (*p* < 0.001; Figure 5C). In contrast, in the OLE-treated EAE mice, both the MDA and AOPP levels were remarkably lowered (*p* < 0.001 and *p* < 0.05, respectively), and the FRAP levels were found notably elevated (*p* < 0.01), reaching levels close to those in the normal group. 

### 2.4. Effect of OLE on Microbioma in EAE Mice 

Fecal DNA was isolated from mice of the different experimental groups to check whether OLE treatment could modulate gut microbioma in EAE mice.

16S rDNA sequencing analysis revealed diverse microbial populations in the experimental groups, but no significant differences were detected among them in the assessed α-diversity metrics: Chao1, Shannon and Simpson indexes (Figure 6A).

In Figure 6B are shown the microbioma profiles according to taxonomic classification at different levels. No major differences could be seen in the composition of gut microbiota at the phylum, order, and family levels among mice of the different experimental groups. At the phylum level, *Bacteroidetes* and *Firmicutes* were major phyla in the gut bacteria of all groups with the combination of the two phyla making up approximately 90% of the total community. *Proteobacteria* and *Verrucomiceobia* showed low abundance. Though not statistically significant, the *Firmicutes* abundance dropped approximately 1.4 fold, from 48% in untreated-control mice to 34% in OLE-treated mice. Thus, the ratio *Firmicutes* to *Bacteroidetes* in the OLE-treated groups tended to decrease compared to the control group, as shown in Figure 6C, but no significant difference was found (*p* > 0.05). Some differences were observed at family level among the experimental groups, but only those of *Akkermansiaceae,* which belong to phylum *Verrucomicrobia,* achieved significant relevance. Compared to the untreated groups, the relative abundance of *Akkermansiaceae* was significantly higher in the OLE-treated groups (Figure 6D). The *Bacteroidaceae*, a family in the phylum *Bacteroidota*, was also increased in the OLE-treated mice, though not significantly, when compared to untreated ones (as it can be appreciated in the family graph of Figure 6B).

### 2.5. In Vitro Effects of OLE on Human Intestinal Epithelial Caco-2 Cell Monolayers

Finally, to study whether the protective intestinal effects observed in OLE-treated EAE mice also involved direct actions on cells that are essential for maintaining a functional intestinal barrier, mono-cultures of Caco-2 cells were exposed to OLE. We treated Caco-2 cell monolayers with the oxidants hydrogen peroxide (H_2_O_2_) and tert-butyl hydroperoxide (t-BOOH), as well as with some relevant inflammatory cytokines such as TNFα and IL-1β, which were found to be enhanced in the EAE mice model. Firstly, we demonstrated that the presence of OLE had no significant influence on the viability of Caco-2 cells (Figure 7A). Then, we evaluated the ability of OLE to protect Caco-2 cell monolayers from oxidative stress. H_2_O_2_ and t-BOOH stimulation induced a significant ROS accumulation in Caco-2 cells compared to untreated ones, and OLE pretreatment abolished this response (Figure 7B). 

We also investigated the ability of OLE to regulate the secretion of proinflammatory cytokines, such as IL-8, a crucial inflammatory mediator of intestinal injury that exerts deleterious effects on the intestinal mucosa [24]. As shown in Figure 7C, the exposure of cells to IL-1β led to an increasing secretion of IL-8, whereas the presence of OLE inhibited this up-regulated response in a dose-dependent manner. 

Moreover, we studied the effect of OLE on Caco-2 cells, and epithelial barrier function was assessed by measurements of TEER and FD-40 permeability (Figure 7D). The presence of OLE at 5 and 10 µM did not affect Caco-2 epithelial barrier function. However, TNFα stimulation induced a significant decrease in TEER and a significant increase in FD-40 permeability on Caco-2 cells. As expected, cell pre-treatment with OLE attenuated the epithelial barrier dysfunction induced by TNFα.

## 3. Discussion

Growing evidence supports the role of the gut-brain axis in MS pathogenesis [5,6,7]. Therefore, it is interesting to explore new therapeutic strategies that can restore/prevent intestinal alterations in MS and its preclinical model, EAE. Intestinal barrier integrity is essential for the maintenance of intestinal health and homeostasis of internal environment [25]. We had previously observed that EAE causes inflammation and oxidative stress in the intestine of mice, resulting in tissue injury and increased intestinal permeability [11]. OLE, a natural antioxidant and anti-inflammatory active substance, has been shown to be effective in the prevention and treatment of EAE, and we have now demonstrated the capability of OLE to prevent barrier defects and inflammation, as well as oxidative stress, in intestinal tissue of EAE mice. In line with the in vivo data, OLE suppressed IL-1β-induced inflammatory IL-8 production, as well as TNFα-induced barrier dysfunction in human intestinal Caco-2 cell monolayers. Therefore, our findings shed new light on the beneficial role of OLE on intestinal homeostasis, specifically in pathologies, such as EAE/MS, but also with potential use in other diseases associated with intestinal barrier dysfunction [26].

Some gastrointestinal diseases, where the intestinal barrier is impaired, have also been associated with CNS demyelination [27]. Although a causal link between intestinal barrier breakdown and CNS demyelination cannot be concluded with certainty in these cases, there appears to be an association not solely explained by their shared epidemiological and immunological characteristics. The association between these entities is certainly complex and requires further study.

OLE is a secoiridoid derivative present in EVOO. Recently, several studies have also highlighted the positive EVOO actions on gut health. Although specific beneficial effects of some of its phenolic compounds, such as hydroxytyrosol, tyrosol, and oleuropein, have already been examined at the intestinal level, OLE effectiveness on protecting intestinal barrier has not yet been investigated [27,28,29,30,31]. Currently, it has been described that the small intestine plays an important role in OLE absorption, and herein, our study demonstrates the beneficial effect of OLE mitigating gut damage in EAE mice, but it does not address the presence of OLE in intestinal tissue [32,33]. Further research to investigate this point is needed, in order to clarify whether the biological effects attributed to it are due to OLE itself or its biotransformed metabolites.

Although OLE is an ingredient of olive oil from which it has been isolated in the past, a recent new method permitted its isolation from olive leaves [14]. In the current application of the new isolation method, we used olive leaves with elevated oleuropein content, which was used as precursor molecule for the production of oleacein during the extraction procedure. For this purpose, and after the screening of olive leaves from several different varieties, we identified a population of wild olive trees that showed high oleuropein content, which were used as the starting material for the production of OLE with yield > 1% *w*/*w* of dried leaves. The currently applied method permitted the isolation of OLE through a selective extraction procedure without the need for expensive and laborious chromatographic methods. 

In many chronic inflammatory diseases afflicting humankind, both gastrointestinal and non-gastrointestinal increase in intestinal permeability is pointed out as a key element, by allowing the entrance of pathogenic components into the lamina propria and later on into systemic circulation. In MS patients, and likewise in its preclinical models, an increased gut permeability has been widely reported [5,6,10,34]. In MOG_35-55_-induced EAE in C57BL/6 mice, the disease is characterized by intestinal barrier disruptions and the presence of flawed goblet cells [10,11]. In this study, we found that OLE effectively exerted an inhibitory effect on these alterations, showing a reduced FITC-dextran translocation using the ex vivo non-everted gut sac model. Accordingly, Caco-2 cells, stimulated with TNFα to induce cell monolayer permeability, confirmed the protective effect of OLE. TNFα is an inflammatory cytokine found notably elevated in colon from EAE mice, which, in intestinal Caco-2 cells, decreases the transepithelial electrical resistance (TEER) while increasing FITC-dextran permeation [35]. Therefore, we confirm that OLE pretreatment was effective in maintaining the Caco-2 cell monolayer integrity, proving its ability to maintain the integrity of the intestinal barrier.

In addition to the permeability analysis, variations in leaky gut-related markers were also evaluated in serum samples. In EAE mice, a previous study by our group reported that the serum levels of surrogate biomarkers of these events, sCD14 and iFABP, augmented significantly compared to control mice. I-FABP is an intracellular protein specifically expressed in enterocytes and it is released into the circulation when intestinal mucosal damage occurs; and sCD14 is a soluble LPS co-receptor released from monocytes in response to bacterial translocation into plasma, also a sign of an active inflammatory response [36,37,38,39]. In the present study, we observed that EAE-increased levels of these markers were significantly attenuated by OLE treatment. 

Additionally, we found that OLE treatment also diminished EAE-induced goblet cell damage in colon. Goblet cells are mucin-secreting epithelial cells that play vital roles in sustaining the intestinal mucosal barrier [40]. Clinical observations and data from experimental animal models have reported the presence of defective goblet cells and a reduced production of mucosal barrier-related molecules, as critical factors in the triggering of the disorders affecting the gastrointestinal tract [41,42,43].. It is interesting to note that OLE administration to EAE mice not only preserved goblet cells mucins in intestinal tissues, but it was also able to prevent the reduction in galectin-3 levels observed in the colon of untreated mice. Since cell surface-associated mucins form strong complexes with galectin-3, preserving the integrity of the mucosal barrier, these results show how OLE might be contributing to the restoration of the epithelial barrier in EAE [44].

We also found that the expression levels of the neurotrophic factor GDNF were down-regulated in the colon of EAE mice, but when daily treated with OLE, levels were restored. A similar expression pattern was observed in serum samples. Although GDNF is mainly secreted by enteric glial cells, another crucial component of the intestinal mucosal defense system also enterocytes synthesize significant amounts of this neurotrophic factor [45]. GDNF has protective roles on barrier functions by modulating its maturation, as well as the proliferation and apoptosis of the intestinal epithelial cell [23,46]. Reduced levels of GDNF lead to morphological and functional abnormalities of the intestinal barrier function, both in patients with intestinal diseases and in preclinical models [47,48]. Our findings in the EAE model are consistent with these considerations. Interestingly, some phytochemicals, such as polyphenols, have shown neurotrophic factor-like activity by binding to neurotrophic factor receptors; future research should unravel whether OLE also possesses neurotrophic functions through a direct agonistic effect on these receptors [49,50].

A compromised intestinal barrier function has unequivocally been associated with inflammatory conditions in the gut [51,52]. In accordance, high levels of the inflammatory TNFα and IL-1β were observed in the colon of EAE mice, compared to control. 

An increased presence of pro-inflammatory cytokines has been observed in patients and preclinical models of intestinal diseases, and treatments suppressing the inflammatory response alleviated the intestinal dysfunction [53,54,55,56,57,58]. Consistent with these data, the current study demonstrated that OLE treatment decreased the pro-inflammatory cytokine levels in the colon of EAE mice. In addition, the immunoregulatory cytokines, IL-33 and IL-25, were preserved in the colon of OLE-treated EAE mice. A protective function has already been described for these cytokines on mucosal tissue. In line with our observations, decreased IL-25 levels have been found in the intestine of both IBD patients and preclinical models, and IL-25 treatment inhibits experimental intestinal damage in mice [59,60]. Regarding IL-33, this cytokine is associated to goblet cells proliferation and mucin expression, hence, it promotes epithelial integrity and restoration of intestinal homeostasis [61,62]. Moreover, IL-33 and IL-25 possess the ability to influence innate and adaptive immunity, promoting protective Th2 cytokine-mediated responses [59]. In EAE, deviation of the immune system towards a Th2 response correlates with disease resistance. Accordingly, treatment with either IL-25 or IL-33 protects mice from EAE diseases, whereas its deficiency or blockade results in an accelerated/exacerbated EAE phenotype [63,64,65,66,67]. Thus, taking together these reports, preserving the expression of these cytokines may be valuable for OLE-induced protection to EAE mice. Although precise molecular mechanisms involved in these protective actions of OLE have not been addressed in this study, regulation of CD14/TLR4/CD14/MyD88 axis, JAK/STAT, MAPKs and inflammasome, as well as post-translational modification in histone H3 are signaling mediators modulated by OLE that deserve further investigation in the EAE context [18,68]. Since oxidative stress and inflammation is a feedback, another essential factor in the pathogenesis of gastrointestinal mucosal diseases is ROS over-accumulation [69,70]. The amelioration of the oxidative response exerted by OLE in gut tissues was also evidenced, when oxidative stress markers were evaluated. In MS patients a reduced antioxidant capacity has been observed [71]. Accordingly, in EAE mice tissues, including gut, oxygen radicals are overproduced, shifting the endogenous oxidant/antioxidant balance, which is consistent with our present findings [11]. Some of the beneficial effects of OLE might be ascribed to its strong antioxidant capacities [72,73,74] and, herein, it has been demonstrated that OLE significantly reduced ROS accumulation in the colon of EAE mice, as well as the levels of MDA and AOPP indicating that OLE lowered the degree of lipid and protein oxidation to suppress intestinal oxidative stress. In consonance, FRAP values, which reflect the overall redox status, were increased in colon of EAE mice treated with OLE, highlighting its protective antioxidant effect. A direct protective effect of OLE against oxidative stress was also observed in Caco-2 cells, after these cells were incubated with OLE, which significantly attenuated ROS accumulation following exposure to the stressors, hydrogen peroxide or tert-butyl hydroperoxide. Therefore, the antioxidant activity of OLE contributes to the above-mentioned gut-integrity strengthening effect.

Another disturbance we observed in the gut of EAE mice was an increase in the cecal index and OLE treatment prevented this enhancement. Usually, full cecum increases are associated to increased fermentation, consequently, to changes in microbial metabolism. Further investigations are required to elucidate whether these observed changes are due to alterations in the cecal microbe composition or in the bacterial enzymatic expression and activity [75]. Likewise, regarding the protective effect of OLE, its direct impact on cecal microbioma should be considered, as well as its actions on proteins involved in microbial activities [76]

Studies on MS patients and the EAE mouse model suggest that the gut microbiome plays a significant role in both disease progression and severity [77,78,79]. In our experimental condition to examine the effects of OLE administration on mice gut microbiota, specific changes in the abundance of *Akkermansiaceae*, a family of mucin-degrading bacteria, were detected in fecal samples from mice treated with OLE, EAE-induced or not. Interestingly, the bacteria *Akkermansia muciniphila* (the better studied member of the *Akkermansiaceae* family with epithelium remodeling properties) positively correlates with mucus layer thickness and intestinal barrier integrity, and promotes the development of host innate and adaptive immune systems with anti-inflammatory effects [80]. Decreased contents of these bacteria in the intestine are associated with the development of several intestinal diseases, whereas increasing its proportion in gut microbiota through dietary modification or pharmacological intervention has beneficial effects in host health. In our study, it might suggest that OLE increasing the *Akkermansiaceae* family abundance favors a protective gut environment. In line, recent studies have suggested that dietary polyphenols play a role in the modulation of the gut microbiota that may favor positive outcomes [81]. Thus, polyphenols from black tea, red wine grape extract/grape pomace extract or cranberry extract stimulate the growth of beneficial bacteria in the gut microbiota such as *Akkermansia*, belonging to the *Verrumicrobiota* phylum [81]. Although the exact mechanisms of action have not yet been fully established, differences in susceptibility between bacterial groups may depend on resistance to any of the mechanisms by which polyphenols interact with bacteria [82], e.g., through short-chain fatty acids production, which stimulates the goblet cells to produce more mucus to preserve intestinal barrier integrity. Focusing on the olive bioactive constituents, a recent study has shown that the diminished abundance observed in bacteria belonging to the phyla *Bacteriodetes* and *Verrumicrobiota* in mice fed with a high fat diet, were also restored by treatment with an olive leaf extract [83]. Moreover, the authors point that this microbiota restauration is associated with the improvement of the gut barrier function, as well as with the beneficial effect induced by the extract on the metabolic and vascular alterations associated to obesity. In this study, we found that the OLE-mediated increase in the relative abundance of the *Akkermansiaceae family* paralelled the prevention of the intestinal barrier damage induced in EAE mice. However, a causal relationship should be stated by the oral administration of these bacteria (i.e., as a probiotic), or a family member such as Akkermansia muciniphila, in future studies.

Additionally, in our study we noted a trend toward an increase in *Bacteroidaceae* family in mice of OLE-treated groups, but those changes were not significant. Interestingly, other investigations, in EAE mice and MS patients, reported that a dietary intervention that confers protection in the EAE model and improves the disability status scale of the disease on MS patients, also significantly increased the *Bacteroidaceae* family richness [77,78]. Therefore, our findings from the OLE-treated mice are worthy of attention, it being possible that using a higher dose of OLE or a different administration route, the changes in *Bacteroidaceae* family richness could achieve a significant impact. We should also consider that in our experimental design, mice were sacrificed at a time where perturbations to the gut microbial populations have not reached a maximum. Additional work should be performed to test these possibilities. 

## 4. Materials and Methods

### 4.1. Disease Induction and Treatment

Female 8 to 10-week-old C57BL/J6 mice (from Charles River Laboratories, Barcelona, Spain) were housed in the animal care facility at the Medical School of the University of Valladolid and provided with food and water ad libitum. All animal care and experimental protocols were reviewed and approved by the Animal Ethics Committee of the University of Valladolid (3008787) and complied with the European Communities directive 86/609/ECC and Spanish legislation (BOE 252/34367-91, 2005) regulating animal research. 

EAE was induced according to our previous study [21]. EAE-immunized mice received an intraperitoneal injection with vehicle control (DMSO/saline, n = 9) or 10 mg/kg/day of OLE (n = 9) starting from immunization day until the end of the experiment, when untreated EAE mice showed hind limb paralysis (about day 24 post-immunization). Control mice (without EAE induction) were also injected daily with OLE (n = 9) or vehicle control (n = 9) for an equivalent timeframe. OLE was dissolved in normal saline containing 5% DMSO. Animals were monitored blindly and daily by two independent observers and neurological signs were assessed on a scale of 0 to 5, with 0.5 points for intermediate clinical findings as previously described: grade 0, no abnormality; grade 0.5, partial loss/reduced tail tone, assessed by inability to curl the distal end of the tail; grade 1, tail atony; grade 1.5, slightly/moderately clumsy gait, impaired righting ability or combination; grade 2, hind limb weakness; grade 2.5, partial hind limb paralysis; grade 3, complete hind limb paralysis; grade 3.5, complete hind limb paralysis and fore limb weakness; grade 4, tetraplegic; grade 5, moribund state or death, [11,22]. Blood and intestinal sections were collected. Tissues were frozen at −80 °C for protein studies or fixed in 4% paraformaldehyde in PBS, followed by paraffin embedding or OCT embedding then frozen.

### 4.2. Oleacein Isolation

Fresh olive leaves (2 kg) were collected from wild trees growing in Volvi Estate, the largest compact population of wild olive trees in Northern Greece. The leaves were manually separated by the stems and air-dried at room temperature for 10 days until the moisture content was less than <10% (*w*/*w*). Then the intact leaves were mixed with water (10 L) at 25 °C and cut into small pieces in the presence of water using a blender. The mixture remained at 25 °C for 30 min and then it was filtered. The aqueous phase was collected and extracted with dichloromethane (5 L). The organic phase was collected and evaporated using a rotary evaporator under reduced pressure affording a viscous liquid containing oleacein (14 g, purity 95% (*w*/*w*)) with NMR data in accordance with those previously described [14]. In Figure 1A the oleacein structure is shown.

### 4.3. Ex Vivo Intestinal Permeability Assay

Ex vivo detection of intestinal permeability was performed using “intestinal sacs” and following the protocol of Zhong et al. with some modifications [84]. Colon tissue samples were extracted into Krebs-Henseleit bicarbonate buffer (KHBB) containing 8.4 mM HEPES, 119 mM NaCl, 4.7 mM KCl, 1.2 mM MgSO_4_, 1.2 mM KH_2_PO_4_, 25 mM NaHCO_3_, 2.5 mM CaCl_2_, and 11 mM glucose (pH 7.4). Then, one end was sutured and from the other end 100 μL of fluorescein-labeled dextran-40 (FD-40, MW 40 kDa, 10 mg/mL) was injected using a gavage needle, and tied to form a 5 cm sac. After a quick dip in KHBB to remove the presence of fluorophore on the outside, the intestinal sac was incubated in 2 mL of new buffer, at 37 °C for 20 min. Finally, the fluorescence of the FD-40 transferred from the intestinal lumen to the incubation solution (Ex./Em. 485/530 nm) was measured in a fluorimeter. Intestinal permeability was expressed in micrograms of extravasated FD-40/cm/min.

### 4.4. Histological Studies

For histological analysis, mouse colons were fixed in 4% paraformaldehyde, processed and embedded in paraffin. 5-μm-thick tissue sections were stained with Periodic acid-Schiff (PAS) and Alcian blue (AB) (Sigma-Aldrich) to stain general intestinal carbohydrate moieties. Acidic mucins stain blue with AB (pH 2.5), neutral mucins stain pink with PAS, and mixtures of neutral and acidic mucins appear purple. The sections from all experimental groups were stained in one single batch to ensure that differences in the staining pattern were not due to technical manipulations, thereby allowing the comparability of the different samples. The evaluation was performed, in a blinded fashion, in each specimen to control the changes that occurred along the treatment. Histopathological examination was performed with a Nikon Eclipse 90i (Nikon Instruments Inc., Amstelveen, The Netherlands). For quantitative analysis, images were acquired from at least three random fields of view per slice and processed using the ImageJ image analysis program (NIH, Bethesda, MD, USA). The area AB/PAS positive was identified as the ratio to the total tissue area

### 4.5. Analysis of Superoxide Anion Production in Colon from EAE Mice

Colon segments were collected and frozen immediately in Tissue-Tek O.C.T. To evaluate the intracellular superoxide anion (O_2_·^−^) the oxidative fluorescent dye dihydroethidium (DHE, Invitrogen Life Technologies, Burlington, Canada), was used as previously described [11]. Briefly, frozen samples cut into 12-μm thickness sections using a cryostat were equilibrated in Krebs-HEPES buffer (NaCl 130 mM, KCl 5.6 mM, CaCl_2_ 2 mM, MgCl_2_ 0.24 mM, HEPES 8.3 mM, glucose 11 mM, pH 7.4) in a humidified and light-protected chamber at 37 °C. The sections were then incubated with 5 μM of DHE for 30 min at 37 °C. Fluorescence signals were viewed using a fluorescence microscope (Nikon TE2000, Japan) under a 10× objective (100× final magnification) and a 20× objective (200× final magnification).

At least five images of each colon sample were captured for analysis using a fixed exposure time for all groups. The intensity of fluorescence signals was quantified using ImageJ software (NIH, Bethesda, MD, USA). A single researcher who was unaware of the experimental groups performed the analysis.

### 4.6. Inflammatory Markers on EAE Mouse Samples Using an Enzyme-Linked Immunosorbent Assay (ELISA)

Serum and colon tissue samples were collected from animals on day 24 after immunization. Colon tissues were weighed and homogenized (1:10, *w*/*v*) in ice-cold PBS supplemented with 0.4 M NaCl, 0.05% Tween 20, 1% EDTA and a protease inhibitor cocktail containing PMSF, leupeptin and aprotinin (Sigma-Aldrich, St Louis, MO, USA), and centrifuged at 10.000 rpm for 10 min at 4 °C. All samples were immediately stored at −80 °C. Mouse TNFα, IL-1β, pro-IL-1β, and IL-25 ELISA kits from eBioscience (San Diego, CA, USA). Mouse sCD14, IL-33 and Galectin-3 (Gal-3) DuoSet ELISA Kits were from R&D (R&D Systems, Minneapolis, MN, USA). Mouse iFABP and GDNF ELISA kit were from Cusabio (Cusabio Biotech Co., Ltd., Wuhan, China). Mice n = 5–7 per group. 

### 4.7. Analysis of Ferric Reducing Antioxidant Power (FRAP) of Colon

Colon homogenate samples were used to measure the total antioxidant activity using the FRAP assay following the method described by Benzie and Strain [85]. FRAP values were calculated according to the calibration curve for FeSO4·7H2O and expressed as µM of Fe^2+^ equivalents. 

### 4.8. Determination of Malondialdehyde (MDA)

Colon homogenate samples were assessed in duplicates to determine the presence of lipid peroxidation products as malondialdehyde (MDA) concentration. The lipid peroxidation level was measured spectrophotometrically by the estimation of MDA concentration based on the reaction with thiobarbituric acid [86]. Briefly, colon supernatans were added to a reaction mixture consisting of 0.373% thiobarbituric acid, 15% trichloroacetic acid and 0.015% BHT. Then, the mixture was heated at 95 °C for 40 min, and cleared by centrifugation at 3.800 rpm for 10 min. The absorbance was measured at 532 nm using a 96-well plate. 

### 4.9. Determination of Advanced Oxidation Protein Products (AOPP)

Colon homogenate samples were assessed in duplicates to determine the presence of advanced oxidation protein products (AOPP) as a biomarker of oxidative stress. 20 µL colon supernatant samples were pipetted into a 96-well microplate and diluted into 100 µL in PBS. Then, 10 µL of 1.16 M KI, and 20 µL absolute acetic acid were added to each well of the microtiter plate. The absorbance of the reaction mixture was immediately read at 340 nm on the VERSAmax microplate reader against a blank containing 100 µL PBS, 20 µL acetic acid, and 10 µL KI solution. AOPP were calibrated with a chloramine-T solution (0–100 µM) that absorbs at 340 nm in the presence of 10 µL of 1.16 M potassium iodide. AOPP concentrations were expressed as µM chloramine-T equivalents.

### 4.10. Microbiota Analysis

Bacterial DNA was extracted from 220 mg of the fecal content from each animal using QIAamp Fast DNA Stool Mini Kit (Qiagen, QIAGEN Iberia, S.L. Spain) according to manufacturer’s instructions with prior disruption using silica beds in a Fastprep^®^ device (QBiogene, Carlsbad, CA, USA). The DNA concentration was determined using a Qubit^®^ fluorimeter (Invitrogen, Waltham, MA, USA). Microbial diversity was studied by sequencing the amplified V3–V4 region of the 16S rRNA gene using previously reported primers and PCR conditions [87]. Sample multiplexing, library purification, and sequencing were carried out as described in the “16S Metagenomic Sequencing Library Preparation” guide by Illumina. Libraries were sequenced on a MiSeq platform, leading to 300-bp, paired-end reads. Demultiplexed fastq files were processed using QIIME2 ™ pipeline version 2022.8 for quality filtering of the reads, merging of the paired ends, chimera removal, and assignation of amplicon sequence variants (ASV) [88]. 

### 4.11. In Vitro Studies

*Cell culture:* Human Caco-2 cells (kindly provided by Dr. E. Arranz, IBGB-UVa/CSIC, Spain) were routinely maintained in DMEM (glutamine, high glucose), supplemented with 10% FCS, 100 U/mL penicillin and 100 pg/mL streptomycin (Life Technologies, Carlsbad, CA, USA), and were incubated at 37 °C in 5% CO_2_. Medium was changed every 2 days and cells were used between passage 19 and 35. The monoculture of Caco-2 cells formed tight junctions at day 17–21 post-confluence. Differentiated cell layers showing high transepithelial resistance (TEER) values (~400–500 Ω × cm^2^), were measured with Millicell electrodes (Millicell-ERS, Millipore, Billerica, MA, USA). 

*Viability assay:* Cell viability was evaluated by using the Promega kit (Madison, WI, USA), Cell Titer 96^®^ Aqueous One Solution Cell Proliferation Assay, according to the manufacturer’s recommendations. Briefly, Caco-2 cells were seeded in 96-well plates (10 × 10^3^ cells/well) and serum starved for 24 h. Then, cells were incubated in the presence of different doses of OLE. After 24 h of incubation, formazan product formation was assayed by recording the absorbance at 490 nm in a 96-well plate reader (OD value). Formazan is measured as an assessment of the number of metabolically active cells. Three different assays were each performed in triplicate.

*Cytokines analysis:* Supernatants of Caco-2 cells stimulated with 25 ng/mL of IL-1β for 48 h in the presence of different doses of OLE were used to quantify IL-8 production. The specific human IL-8 ELISA Ready-Set-Go kit (eBioscience, San Diego, CA, USA) was used according to the manufacturer’s protocol. 

*Measurement of intracellular reactive oxygen species (ROS) levels:* ROS levels were measured with the probe dichlorodihydrofluorescein diacetate (DCFH-DA; Molecular Probes, Eugene, OR). Briefly, Caco-2 cells were seeded in 96-well microplates at 1 × 10^4^/well and after serum starvation, cells were incubated overnight at 37 °C with the indicated doses of OLE. Then, cells were loaded with 10 μM of DCFH-DA for 30 min at 37 °C. After that, cells were stimulated with 500 μM of H_2_O_2_ or 400 μM of tert-butyl hydroperoxide (t-BOOH) for 1 h. The fluorescent signal was measured at Ex. 485 nm-Em. 530 nm, using a plate reader Fluoroskan Ascent FL (Thermo Electron Corporation, Waltham, MA, USA). Results were expressed as an n-fold increase over the values of the control group.

*Transepithelial electrical resistance (TEER) measurement:* The integrity of a Caco-2 monolayer was determined by measuring the TEER value [89]. Cells were grown in 24-well plates and seeded at 1 × 10^5^ cells/insert onto polycarbonate membrane Transwell inserts with 0.4 μm pore size, and 0.33 cm^2^ growth surface (Corning, Inc.; Lowell, MA, USA). Cells were cultured for 21 days to reach differentiation. After that, Caco-2 cell monolayers were pretreated with the indicated doses of OLE for 30 min (apical) and then stimulated with 100 ng/mL of TNFα for 24 h at 37 °C. TEER values were measured with Millicell electrodes (Millicell-ERS, Millipore, Billerica, MA, USA). TEER recorded in unseeded Transwell inserts was subtracted from all values. TEER measures were normalized to untreated-control cell and expressed as a percentage of control.

*Permeability studies:* Permeability of the cell monolayer was determined by using the macromolecular tracer FITC-labeled Dextran (FD-40, Sigma Chemical Co. St. Louis, MO, USA). Confluent and differentiated Caco-2 cell monolayers were pretreated with the indicated doses of OLE for 30 min (apical) and then stimulated with 100 ng/mL of TNFα for 24 h. Then, the medium was aspirated and both chambers were washed with HBSS. After that, 200 μL of 10 mg/mL FITC-dextran dissolved in HBSS was added at the apical compartment of each insert. After 1 h incubation at 37 °C, 200 μL aliquots were taken from the basolateral chamber and plated into a black, flat-bottom 96-well plate. The fluorescence intensity was measured in a Fluoroskan Ascent FL microplate reader (Thermo Electron. Corporation, Waltham, MA, USA) with the setting of Ex. 485 nm and Em. 530 nm. The amount of FITC-Dextran transported into the basolateral compartment (permeability flux) was extrapolated from a standard curve and expressed as mg/mL/h. Results were expressed as apparent permeability coefficient (Papp) and defined as cm/h. “Papp” is derived from the ratio of flux rate (mg/mL/h) to that of initial concentration (in mg/mL) and surface area of the membrane.

### 4.12. Statistical Analyses

Data analyses were performed using one-way ANOVA (for multiple comparisons) or two-way ANOVA (for four-groups comparisons). The Bonferroni test was utilized for post hoc analysis among multiple groups where appropriate. Results described as mean ± SD. *p* < 0.05 were considered statistically significant. Statistical analyses were performed using the GraphPad Prism Version 4 software (San Diego, CA, USA).

## 5. Conclusions

In conclusion, our findings demonstrate for the first time that OLE effectively regulates intestinal oxidative stress, inflammation, and permeability when administered to EAE mice. Since OLE also ameliorates MS classical clinical signs in EAE, this study remarks the probable relevance of the intestinal alterations in the evolution of the disease. Additional studies are necessary to check whether OLE can be used to improve MS and MS-related disorders in patients, but our data strongly support the therapeutic potential of OLE for the treatment of gastrointestinal diseases where the intestinal barrier is impaired, including those associated with CNS demyelination.

## Figures and Tables

**Figure 1 ijms-24-04977-f001:**
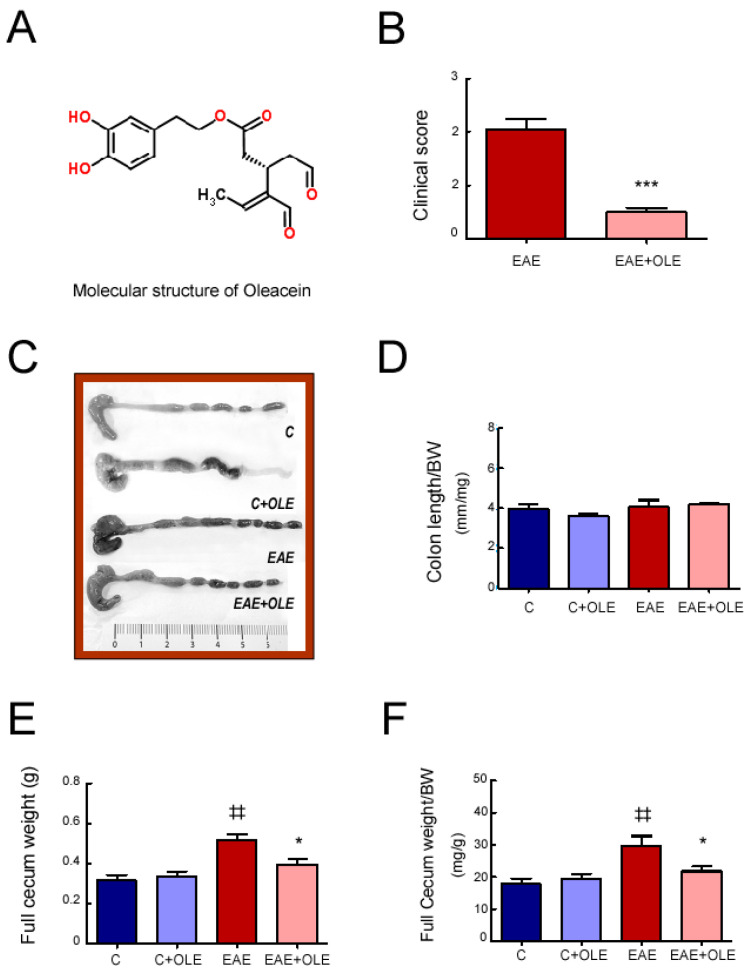
**Effect of OLE treatment on colon length and cecal parameters in EAE mice.** (**A**) Schematic representation of OLE; (**B**) Maximal scores; (**C**) Representative image of colons; (**D**) colon length/body weight (BW) ratio; (**E**) full cecum weight; (**F**) cecal index: full cecum weight/body weight (BW) ratio. Results were expressed as the mean ± SEM, n = 5–7 per group. ^‡‡^ *p* < 0.01 vs. control; and * *p* < 0.05 and *** *p* < 0.001 vs. untreated-EAE. C, healthy mice. C + OLE, healthy mice treated with OLE. EAE, induced mice. EAE + OLE, induced mice treated with OLE.

**Figure 2 ijms-24-04977-f002:**
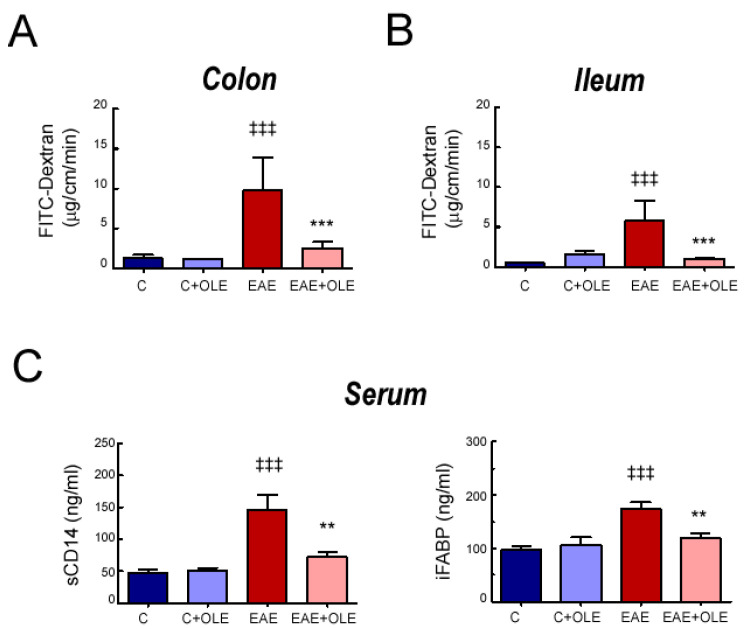
**OLE treatment protects from intestinal permeability in EAE mice.** Intestinal sacs prepared from colon (**A**) and Ileum (**B**) to assess intestinal permeability. Sacs were loaded with FITC-labeled Dextran (FD-40) and placed in a bath. After 120 min, the FD-40 concentration from the bath solutions was quantified. (**C**) Levels of soluble CD14 and iFABP in serum, quantified by ELISA. Results were expressed as the mean ± SEM, n = 5–7 per group. ^‡‡‡^
*p* < 0.001 vs. control; and ** *p* < 0.01 and *** *p* < 0.001 vs. untreated-EAE. C, healthy mice. C + OLE, healthy mice treated with OLE. EAE, induced mice. EAE + OLE, induced mice treated with OLE.

**Figure 3 ijms-24-04977-f003:**
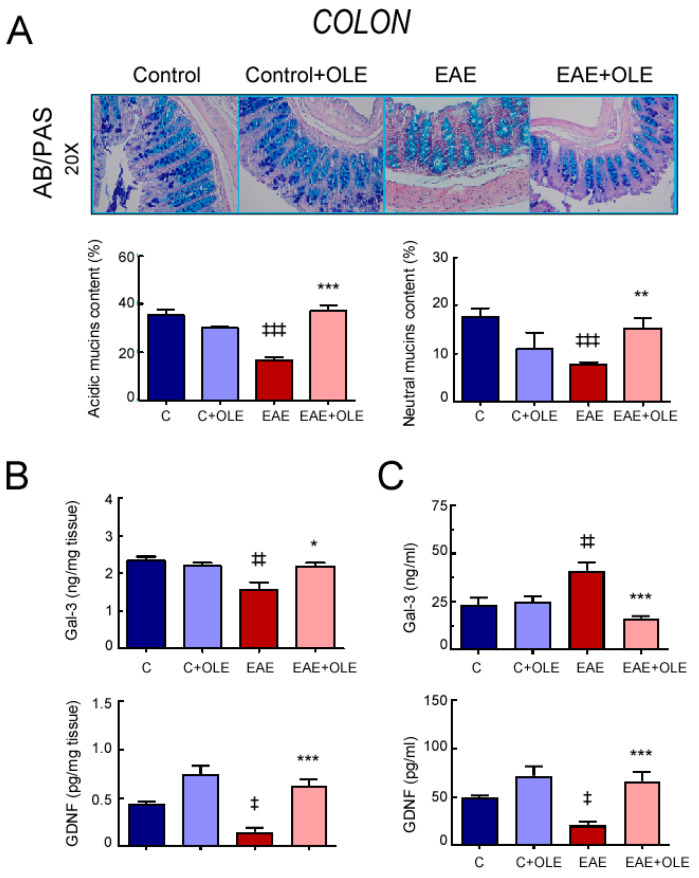
**OLE treatment protects from intestinal mucin alteration in EAE mice.** (**A**) Histological analysis of colon mucins stained with Alcian Blue/Periodic acid-Schiff (AB/PAS). Objective lens 10×. Representative photomicrographs and quantification graphs. Expression of galectin-3 (Gal-3) and glial cell line-derived neurotrophic factor (GDNF) in colon tissue (**B**) and in serum (**C**) were quantified by ELISA. Results were expressed as the mean ± SEM, n = 5–7 per group. ^‡^
*p* < 0.05, ^‡‡^ *p* < 0.01 and ^‡‡‡^ *p* < 0.001 vs. control; and * *p* < 0.05, ** *p* < 0.01 and *** *p* < 0.001, vs. untreated-EAE. C, healthy mice. C + OLE, healthy mice treated with OLE. EAE, induced mice. EAE + OLE, induced-mice treated with OLE.

**Figure 4 ijms-24-04977-f004:**
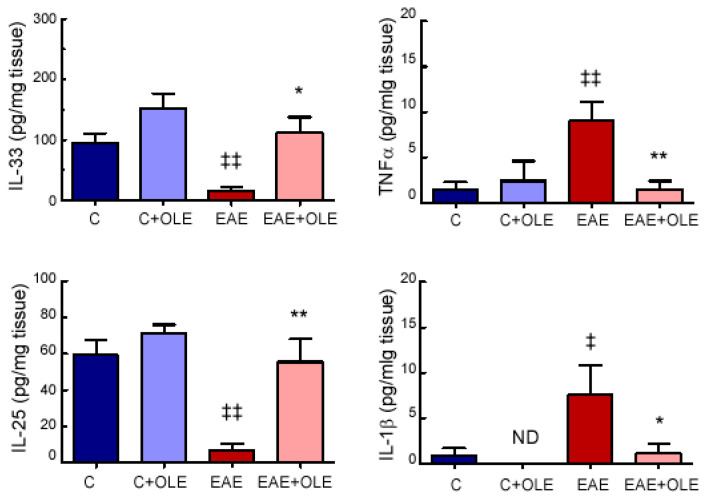
**OLE treatment modulates inflammatory parameters in colon tissue from EAE mice.** Levels in colon of the inflammatory mediators IL-33, TNFα, IL-25 and IL-1β were quantified by ELISAs. Bar graphs represent the mean ± SEM of 5–7 animals. ^‡^
*p* < 0.05 and ^‡‡^
*p* < 0.01 vs. control; and * *p* < 0.05 and ** *p* < 0.01 vs. untreated-EAE. C, healthy mice. C + OLE, healthy mice treated with OLE. EAE, induced mice. EAE + OLE, induced mice treated with OLE.

**Figure 5 ijms-24-04977-f005:**
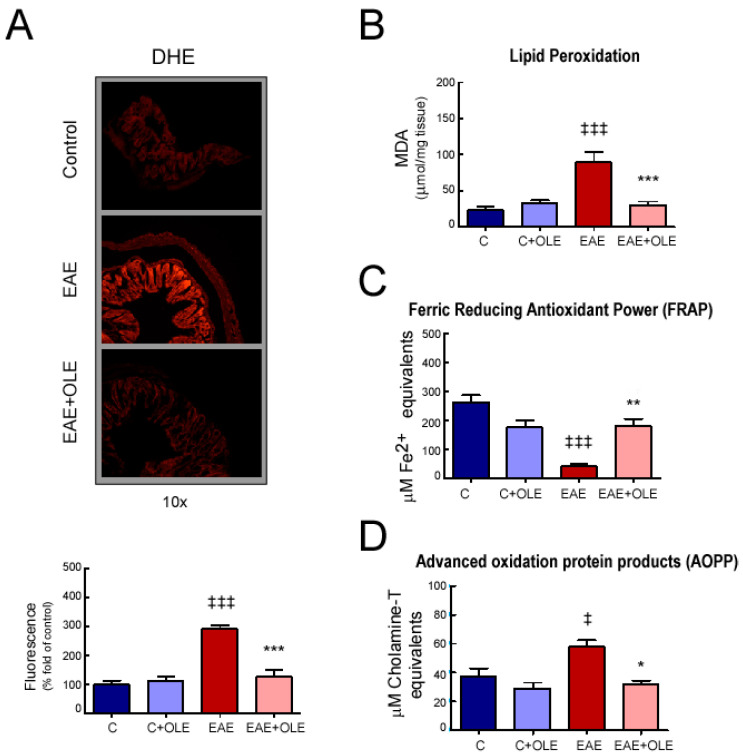
**OLE treatment reduces oxidative stress in colon tissue from EAE mice.** (**A**) Representative photomicrographs of colon tissue stained with DHE. Histological analysis by fluorescence microscopy and quantification. Objective lens 10×. Expression levels of malondialdehyde, MDA (**B**); advanced oxidized protein products, AOPP (**C**); and ferric reducing/antioxidant power, FRAP (**D**) in colon tissue. Results were expressed as the mean ± SEM, n = 4–7 per group. ^‡^
*p* < 0.05 and ^‡‡‡^
*p* < 0.001 vs. control; and * *p* < 0.05, ** *p* < 0.01 and *** *p* < 0.001 vs. untreated-EAE. C, healthy mice. C + OLE, healthy mice treated with OLE. EAE, induced mice. EAE + OLE, induced mice treated with OLE.

**Figure 6 ijms-24-04977-f006:**
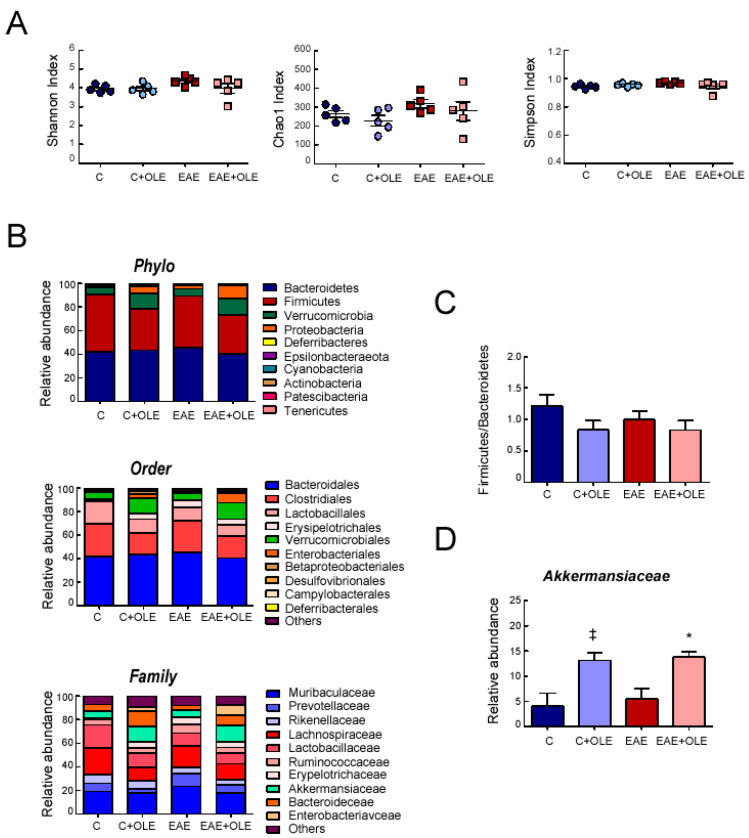
**Effect of OLE treatment on intestinal microbiota of mice**. (**A**) Alpha-diversity of the gut microbiota using Shannon, Simpson and Chao indices. (**B**) Bar chart regarding the distribution of the most abundant phyla, orders, and families. The proportion of stack in the bar chart corresponds to the total amount of reads of the most abundant phyla, orders and families. (**C**) *Firmucutes*/*Bacteriodetes* ratio (**D**) Relative abundance of *Akkermansiaceae*. Data are the mean ± SEM, n = 5 per group. ^‡^
*p* < 0.05 vs. control; and * *p* < 0.05 vs. untreated EAE. C, healthy mice. C + OLE, healthy mice treated with OLE. EAE, induced mice. EAE + OLE, induced mice treated with OLE.

**Figure 7 ijms-24-04977-f007:**
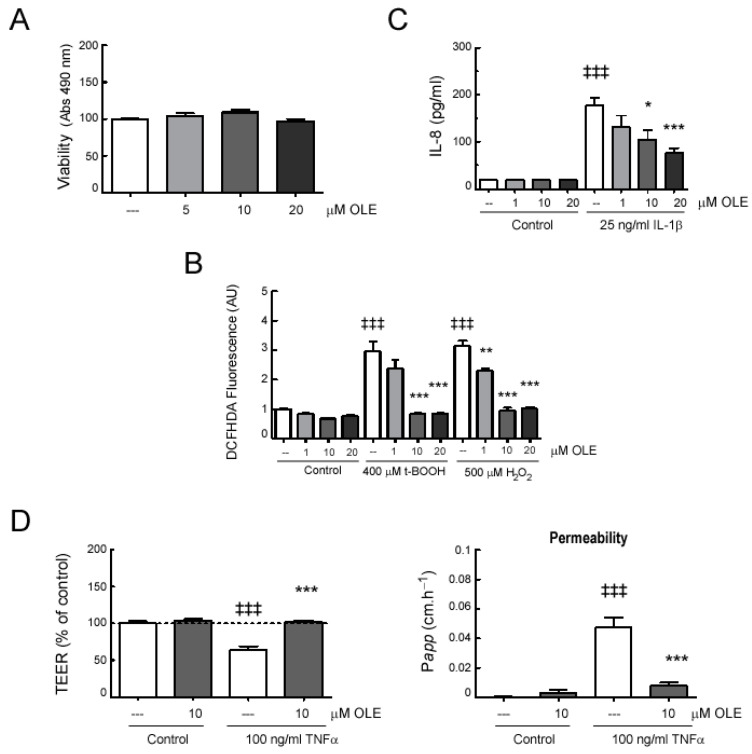
**OLE treatment modulates responses of activated Caco-2 cell monolayers.** (**A**) Caco-2 monolayers were treated for 24 h with the indicated doses of OLE and viability was measured as indicated in material and methods. Caco-2 monolayers, pretreated for 30 min with different doses of OLE, were incubated with the indicated stimuli: (**B**) After 1 h, cells were stained with DCFH-DA and intracellular ROS-production was analyzed in a microplate reader; (**C**) after 24 h, IL-8 concentration in the cell-culture supernatant was measured by commercial ELISA; and (**D**) after 24 h, trans-epithelial electrical resistance (TEER) was tested. TEER values at 24 h normalized to the untreated control (100%) and FITC-dextran (FD-40) transport. Assays were performed in duplicates, n = 3. Results were expressed as the mean ± SEM. ^‡‡‡^
*p* < 0.001 vs. control; and * *p* < 0.05 and ** *p* < 0.01 *** *p* < 0.001 vs. stimuli without OLE.

## Data Availability

Data are contained within the article or Appendix A.

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
