# Peer review of "Treatment with the Olive Secoiridoid Oleacein Protects against the Intestinal Alterations Associated with EAE"

_ijms, 2023, doi:10.3390/ijms24054977_

Round 1

Reviewer 1 Report

The manuscript Treatment with the olive secoiridoid oleacein protects against the intestinal alterations associated with EAE focuses on a very important and interesting topic - EAE-intestinal dysfunction and the impact of OLE treatment on gut barrier protection. Although the research is presented in a very concise and refreshing manner providing valuable info on the OLE effect in EAE-intestinal dysfunction there is one major and few minor issues/points in sections Materials and methods and Results that need to be addressed by the authors.

SECTION MATERIALS AND METHODS

Minor concerns and suggestions

“Animals were monitored blindly and daily by two independent observers and neurological signs were assessed on a scale of 0 to 5, with 0.5 points for intermediate clinical findings as previously described [11,21] ” I believe it is of great importance that the authors describe the assessment of neurological signs i.e. dysfunction in detail since it is crucial for evaluation of success of EAE induction and thus represents the basis of the whole experimental setup.

Major concern

The authors state that the data were analyzed by one-way ANOVA and Bonferroni test for post hoc comparisons among multiple groups where appropriate. Nevertheless, presented experimental setup included 4 groups - C, healthy mice; C+OLE, healthy mice treated with OLE; EAE, induced mice; EAE+OLE, induced-mice treated with OLE. Please provide the explanation for usage of one-way ANOVA instead two-way ANOVA which is more appropriate and would certainly provide more reliable data or revise the results i.e., statistical analyses (replace the one-way with two-way ANOVA).

SECTION RESULTS

I would kindly suggest to the authors to refer only to results presentation in the section Results and shorten this part by omitting the parts of the text not relevant to results presentations/interpretation. For example: Subsection 3.1. OLE treatment protected against EAE-induced intestinal mucosal barrier damage in mice “In a previous research, we demonstrated that OLE administration to MOG35-55 immunized mice protected CNS tissues from inflammatory damage and was sufficient to ameliorate the classical EAE disease characterized by a progressive ascendant paralysis. Recent reports indicate that intestinal permeability is affected during the course of EAE, therefore, we wondered whether EAE mice daily treated with OLE also improved gut dysfunction [10,11]. We therefore examined the effect of OLE on intestinal permeability, barrier function, as well as on intestinal oxidative stress and inflammation. The study was performed in EAE mice at 24 days after EAE induction, when mice of the untreated-EAE group showed one-sided hindlimb paralyses (clinical score 2), at minimum, while, mice in the OLE-treated EAE group showed inability to curl the distal end of the tail (clinical score 0.5) (Figure 1B)… ” Suggestion: delete or shorten the text. The same applies for the other Subsections in this Section.

Reviewer 2 Report

In this study the authors showed protective effects of OLE on EAE involves normalizing the gut alterations associated to the disease. Some concerns and suggestions are listed as below:

In Figure 1, why maximal scores of EAE were 2 in this study? I wonder if OLE has any effects on EAE incidence.

In Figure 4, did you test other cytokines, such as IL-10?

The concentration of OLE in intestinal tissue should be measured.

I wonder if therapeutic effects of OLE are dose-dependent.

Any potential side effects?

In Figure 7, why only IL-8 was measured?

Cellular and molecular mechanisms are lacking in this study. This is a major concern.

What about the targets of OLE?

Reviewer 3 Report

The research article is well-written, the results are thorough and cover the primary goals of the study. However, implementing the following suggestions might improve and enhance the quality of the manuscript (Manuscript ID: IJMS-2149972).

 First, Author has not mentioned line number in the manuscript, it is very difficult to review manuscript. I don’t know what so reason. However, I have reviewed, and the following comments are:

In results section 3.1-page number 7 figure 1C, The representative image of colons, why there is difference in color (less dark color in control, subsequently increases the darkness of colon upon treatment of OLE and why EAE has long colon length than control and treatment of OLE), is there any association of colon color and length with OLE is not clear. Also, in figure 1D, I am wondering whether colon length increases in EAE than control and whether treatment of OLE is statistically significant is not clear.

In figure 1E and F, Author showed cecal that the full cecal weight, as well as the ration full cecum weight/body weight, were higher in EAE mice, and OLE treatment prevented this increase (Figure 1E and F). What is possible explanation why OLE treatment reverses this process.

 In figure 2A and B, why author used same statistically signs (* and ‡ are <0.05) and what is the difference between * and ‡ as showed both are (* and ‡ are <0.05). Similarly, in figure4, the statistically signs like ‡p<0.05 ‡and ‡p<0.05‡ and ‡p><0.01, is confusing.

Author has shown that number of genes were altered upon treatment of OLE like increased levels of IL-33, IL-25 and decreases TNFa, IL1b, MDA and other genes. However, I am wondering that whether OLE directly or indirectly and how OLE regulates the expression levels is not clearly and has not been mentioned anywhere in the manuscript (discussion part).

Similarly, In figure 6C and D, author showed that the ratio Firmicutes to Bacteroidetes in the OLE-treated groups tended to decrease compared to the control group, but only those of Akkermansiaceae which belong to phylum Verrucomicrobia achieved significant relevance, is there any association with OLE and Akkermansiaceae and how it specifically alters the Akkermansiaceae group whether OLE specially enhances growth of Akkermansiaceae is not mentioned

Author should have cited the recent references like https://doi.org/10.3390/nu14183773 as well.

Round 2

Reviewer 2 Report

The authors have answered my questions.